# DNMT Enzymes and Their Impact on Cervical Cancer: A State-of-the-Art Review

**DOI:** 10.3390/ijms262110496

**Published:** 2025-10-29

**Authors:** Eric Genaro Salmerón-Bárcenas, Andrea Martínez-Zayas, Miguel Vargas-Mejía, Nicolas Villegas-Sepúlveda, Paola Briseño-Díaz, Arturo Aguilar-Rojas, Christian Johana Baños-Hernández, Francisco Israel Torres-Rojas, Ramón Antaño-Arias, Rosaura Hernández-Rivas

**Affiliations:** 1Departamento de Biomedicina Molecular, Centro de Investigación y de Estudios Avanzados del Instituto Politécnico Nacional, Ciudad de Mexico 07360, Mexico; eric.salmeron@cinvestav.mx (E.G.S.-B.); andrea.martinez@cinvestav.mx (A.M.-Z.); mavargas@cinvestav.mx (M.V.-M.); nvillega@cinvestav.mx (N.V.-S.); 2Departamento de Genética y Biología Molecular, Centro de Investigación y de Estudios Avanzados del Instituto Politécnico Nacional, Ciudad de Mexico 07360, Mexico; paola.briseno@cinvestav.mx; 3Unidad de Investigación Médica en Medicina Reproductiva, UMAE en Ginecología y Obstetricia No. 4, Instituto Mexicano del Seguro Social, Ciudad de Mexico 01070, Mexico; a_aguilar@unam.mx; 4Instituto de Investigación en Ciencias Biomédicas, Centro Universitario de Ciencias de la Salud, Universidad de Guadalajara, Guadalajara 44340, Mexico; johana.banos@academicos.udg.mx; 5Laboratorio de Biomedicina Molecular, Facultad de Ciencias Químico Biológicas, Universidad Autónoma de Guerrero, Chilpancingo 39086, Mexico; ftorres@uagro.mx (F.I.T.-R.); ramon.a@uagro.mx (R.A.-A.)

**Keywords:** 5-mC, DNMT1, DNMT2, DNMT3A, DNMT3B, DNMT3L, methylation, cervical cancer

## Abstract

Genomic DNA methylation is an epigenetic modification that primarily occurs at CpG sites and is associated with the transcriptional repression of genes. This process plays a crucial role in maintaining cellular homeostasis and is catalyzed by a family of enzymes known as DNA methyltransferases (DNMTs), which includes DNMT1, DNMT2, DNMT3A, DNMT3B, and DNMT3L. DNMT1 is classified as a maintenance methyltransferase, whereas DNMT3A and DNMT3B are responsible for de novo methylation. Altered expression of DNMTs has been reported in various human diseases, including cancer. Cancer remains a major global health issue, with an estimated 20 million new cases and 9.7 million deaths reported in 2022. Among women, cervical cancer (CC) ranks fourth in both incidence and mortality worldwide, with persistent infection by high-risk human papillomavirus (HR-HPV) being the primary risk factor. Several studies have demonstrated that DNMT expression and activity are upregulated in CC, suggesting their potential as diagnostic and prognostic biomarkers. HR-HPV infection appears to increase DNMT expression, thereby promoting cervical carcinogenesis through aberrant methylation and subsequent silencing of tumor-suppressor genes such as PTEN, PAX1, and TSLC1. Furthermore, DNMTs are being explored as therapeutic targets in CC. In this review, we summarize the current state of the art regarding DNMTs in cervical cancer and discuss their functional roles and potential utility as diagnostic, prognostic, and therapeutic biomarkers.

## 1. Introduction

In eukaryotes, at least 98% of DNA methylation occurs in the CpG dinucleotide context, promoting 5-methylcytosine (5-mC) formation [1]. DNA methylation in CpA, CpT, and CpC dinucleotide contexts (non-CpG methylation), however, has unfortunately been less investigated [2]. 5-mC is mainly located in genomic regions enriched in CpG dinucleotides known as CpG Islands (CGIs) that have a high CG content > 50%, an observed CpG ratio (known as Obs/Exp) > 0.6, and an extent > 200 bps [3]. About 70% of promoters in the human genome contain at least one CGI [4]. Methylation in gene promoters can promote the transcriptional activation of genes, such as *FOXA2* and *PAX2* [5,6], or inhibit the expression of genes, such as *AJAP1* and *KLF2* [7,8]. Similarly, methylation in enhancers and super-enhancers is associated with transcriptional repression [9]. In contrast, methylation in the gene body increases the expression of genes, for example, the *TRIB3* and *STC2* genes [10]. Finally, methylation in CpGs present in repetitive elements, such as tandem and interspersed repeats, is associated with genomic stability [11]. For many years, it was thought that DNA methylation was an irreversible and very stable epigenetic mark [12,13]; however, it is now known that there is active or passive demethylation of 5-mC by the Ten-Eleven-Translocations (TETs) enzyme family (TET1, TET2, and TET3). These enzymes are Fe (II) and 2-OG-dependent methylcytosine dioxygenases and convert 5-mC to 5-hydroxymethylcytosine, 5-formylcytosine, and 5-carboxylcytosine, which is replaced by an unmodified cytosine [14,15,16,17].

DNA methylation is catalyzed by DNA methyltransferases (DNMTs), an enzyme family. In humans, the DNMT family consists of five members: DNMT1, DNMT2, DNMT3A, DNMT3B, and DNMT3L. DNMT1, DNMT3A, and DNMT3B are canonical DNMT enzymes, while DNMT2 and DNMT3L are non-canonical DNMT enzymes [18]. Expression of DNMT enzymes is increased in several cellular processes, such as DNA replication, as well as in several human diseases, including cancer [19,20].

According to the International Agency for Research on Cancer, approximately 20 million new cases of cancer and 9.7 million deaths from cancer occurred worldwide in 2022. Unfortunately, it has been estimated that one in five people will get cancer during their lifetime. In this sense, cervical cancer (CC) is one of the most common cancers in women, with an estimated total of 661,021 new cases and 348,189 deaths worldwide in 2022 [21]. The main risk factor for CC is infection by High-Risk Human Papilloma Virus (HR-HPV), including HR-HPV16 and HR-HPV18; smoking; having first sexual intercourse at a young age; having multiple sexual partners; the use of oral contraceptives; several full-term pregnancies; and infection by Human Immunodeficiency Virus (HIV) [21,22,23]. Moreover, the high incidence and mortality of CC are associated with a medium and low human development index [21]. In 2022, the American Cancer Society reported that CC is ranked fourth and sixth place in incidence and mortality in female cancers, with a total of 14,100 new cases and 4280 new deaths estimated in the United States [24]. In addition, the regions of each continent with the highest incidence and mortality rates from CC are Eastern Africa (incidence of 40.4 and mortality of 28.9 per 100,000 women/year), South-Eastern Asia (incidence of 17.4 and mortality of 9.5 per 100,000 women/year), Eastern Europe (incidence of 15.7 and mortality of 6.3 per 100,000 women/year), and South America (incidence rate of 15.6 and mortality of 7.8 per 100,000 women/year) [21].

It is well known that CC develops from precursor lesions of the cervix, which are classified according to two systems established by the World Health Organization (WHO) and the United States National Cancer Institute. In 1968, the WHO named these precursor lesions Cervical Intraepithelial Neoplasia (CIN), which are categorized into grades CIN1–3 according to the grade of proliferation of atypical basaloid cells. CIN1 indicates mild dysplasia (abnormal growth and development of cells), CIN2 indicates moderate dysplasia, and CIN3 indicates severe dysplasia, which can progress to CC [25,26]. Later, the NCI of the USA established the Bethesda classification system in 1988 and introduced the term Squamous Intraepithelial Lesion (SIL) to describe a precancerous lesion and reclassified the three groups established by WHO into only two groups: named low-grade SIL (LSIL) and high-grade SIL (HSIL) [27,28]. LSIL is characterized by cytologically extensive changes associated with Low- or Intermediate-Risk HPV (LR-HPV or IR-HPV), while the cellular changes in HSIL are related to HR-HPV, and this lesion can advance to CC [25,29]. Currently, the CC diagnostic process is based on cytological screening (Pap smears); non-cytological tests, such as visual inspection using acetic acid (VIA), accompanied by immunohistochemistry assays, such as p16 and ki-67; and molecular screening, which includes HPV in situ hybridization; and a biopsy [30,31,32]. Moreover, there are other biomarkers used to help predict the response to immunotherapy drugs, such as immunotherapeutic markers (PD-L1) [33]. Moreover, a recent study demonstrated that the DNA mismatch repair (MMR) system regulates PD-L1 expression through DNMTs in CC [34].

Several studies have evaluated the role of epigenetic mechanisms and their molecular implications in CC, including DNA methylation, histone modifications, and non-coding RNAs. The most common epigenetic mechanism studied in CC is DNA methylation mediated by DNMT enzymes [35,36]. Therefore, in this review, we provide an overview of the roles of DNMT1, DNMT2, DNMT3A, DNMT3B, and DNMT3L enzymes in CC, focusing on their aberrant expression; regulatory mechanisms; involvement in cellular processes; and potential as diagnostic, prognostic, and therapeutic biomarkers.

## 2. Discovering DNMT Enzymes

Historically, DNA methylation was one of the first epigenetic mechanisms described in key cell events, such as female X inactivation during eukaryotic cell differentiation [37]. Currently, this epigenetic modification is one of the most commonly studied in several research fields, such as aging and cancer [38]. It is well known that DNA methylation mainly occurs in cytosines, giving rise to 5-mC [1].

According to the literature, 5-mC was identified for the first time in nucleic acids of *Tubercle bacillus* through optical assays in 1925 [39]. In 1948, Hotchkiss reported, for the first time, the quantitative isolation of 5-mC from hydrolyzed calf thymus samples using paper chromatography [40]. Later, in 1950, chromatography and ultraviolet spectrometry were used to confirm the presence of 5-mC on DNA in animal and plant genomes [41]. In 1975, it was reported that 5-mC is a heritable key epigenetic mark in animals that regulates gene expression [37,42]. Afterward, in 1988, the first mammalian DNMT enzyme, called DNMT1, was cloned, purified, and characterized from murine erythroleukemia cells [43]. Three years later, it was cloned and only located on chromosome 19, a portion of the human DNMT1 cDNA. Moreover, the results of Northern blot and RT-PCR assays indicated that DNMT1 expression increases in human cancer cells and patients with colonic neoplasia [44]. In 1992, this enzyme was identified, isolated, cloned, mapped, and characterized in human normal tissues. Interestingly, high DNMT1 expression was observed in placental, brain, heart, and lung tissues using Northern blot assays [45]. A year later, an increase in the expression and activity of DNMT1 enzyme was reported using RT-PCR and DNA-MTase Activity Assays in human colon cancer [46]. Subsequently, the DNMT1 gene was isolated, cloned, and characterized in *Arabidopsis thaliana* in 1993 [44]; in COS1 cells derived from chicken in 1995 [45]; and in embryos of the sea urchin *Paracentrotus lividus* in 1996 [47]. A year later, a novel DNMT enzyme, known as DNMT2, was identified, cloned, mapped, and characterized using library screens; nucleic acid blot analysis; and RNase protection assays from mouse embryo cells and human tissue samples [48].

In 1998, two “de novo” DNMT enzymes (named DNMT3A and DNMT3B) were cloned, mapped, and characterized from mouse and human cells. In addition, two shorter polypeptides of DNMT3B were identified from mouse cells [49]. Subsequently, in 1999, it was shown that DNMT3A and DNMT3B enzymes are fundamental for “de novo” methylation and mammalian embryonic development [50]. Simultaneously, the expression profile of DNMT enzymes in human fetal and adult tissues was studied using Northern blot assays; moreover, their chromosomal locations were mapped using FISH assays. Surprisingly, four isoforms of DNMT3B were identified using RT-PCR and sequencing assays. Moreover, overexpression of DNMT1, DNMT3A, and DNMT3B at the mRNA level was observed in bladder, kidney, colon, and pancreas human cancers [51].

Later, in 2000, another novel DNMT gene, called DNMT3L, was identified, isolated, cloned, mapped, and partially characterized from human tissue samples, including ovary, testis, thymus, and fetal thymus tissues [52].

Finally, in 2007, several DNMT3B isoforms were identified as overexpressed in many human cell lines and primary acute leukemia cell samples, as well as other tumor cell lines [53], as shown in Figure 1.

Several recent studies have shown the oncogenic role of DNMT enzymes in several types of human cancers [54].

## 3. Structure of DNMT Genes and Proteins

The *DNMT1* gene is located on 19p13.2 chromosome in humans, contains a CpG island in its promoter (Figure 2A, upper), and transcribes two mRNAs that code for DNMT1: the isoform A (ENST00000359526.9/hDNMT-202) and the isoform B (ENST00000340748.8/hDNMT-201) (Figure 2A, lower). DNMT1 isoform A is transduced from an mRNA of 41 exons (5274 bp) from the reverse strand and encodes a protein of 1616 amino acids (184 kDa) with an average residue weight of 113.247 g/mol, charge of 23.5, isoelectric point of 7.8892, and molecular weight of 184,819.05 g/mol. The isoform B is transduced from an mRNA of 40 exons (5408 bp) from the reverse strand and encodes a protein of 1616 amino acids (183 kDa) with an average residue weight of 113.345 g/mol, charge of 21.5, isoelectric point of 7.7803, and molecular weight of 183,165.23 g/mol [55,56,57,58].

Recently, a total of 52 transcripts were identified through microarrays and sequence-based assays, which were annotated in Ensembl and HAVANA databases [56]. These transcripts are categorized into 24 transcripts with a retained intron: 6 transcripts whose protein-coding sequence (CDS) is not defined, 9 transcripts with nonsense-mediated decay, and 13 transcripts that codify for a protein, including the short isoform previously reported that lacks the first 118 amino acids from the N-terminal region [56,58,59].

The *DNMT2* gene, also known as tRNA aspartic acid methyltransferase 1 (ENST00000377799.8/TRDMT1-203/hDNMT2-203), is located on the 10p12-10p14 chromosome in humans; contains 11 exons; is transcribed from a reverse strand in an mRNA of 12,918 bp (Figure 2B); and encodes a protein of 391 amino acids (45 kDa), with an average residue weight of 114.058 g/mol, charge of −2.5, isoelectric point of 5.9491, and molecular weight of 44,596.59 g/mol [48,56]. A total of 11 new transcripts were annotated in Ensembl and HAVANA databases, including four transcripts with potential to code for proteins, three transcripts with nonsense-mediated decay, and four transcripts whose CDS are not yet defined [56].

The *DNMT3A* gene is located on the 2.p23 chromosome in humans, containing 23 exons and a CpG island in its promoter (Figure 2C, upper). Moreover, this gene transcribes a 2172 bp mRNA from the reverse strand and encodes DNMT3A1, a 912 amino acid protein, known as full-length or canonical (ENST00000321117.10/hDNMT3A-202) (130 kDa) [60], and an isoform named DNMT3A2, which consists of 689 amino acids and lacks 223 amino acids from the N-terminal (ENST00000402667.1/hDNMT3A-205) (Figure 2C, lower) [56,61,62]. DNMT3A1 has a mass of 101.858 kDa with an average residue weight of 111.687 g/mol, charge of 0.0, isoelectric point of 6.5079, and molecular weight of 101,858.32 g/mol. The DNMT3A2 isoform has a mass of 77.817 kDa and an average residue weight of 112.942 g/mol, charge of −0.5, isoelectric point of 6.4392, and molecular weight of 77,816.76 g/mol [55,56,57,58].

The *DNMT3B* gene is on the 20q11.2 chromosome in humans and contains a CpG island in its promoter and 23 exons (Figure 2D, upper). The *DNMT3B* gene transcribes a 4336 bp mRNA from the forward strand (Figure 2D, lower), which encodes a protein of 853 amino acids (95.751 kDa) (ENST00000328111.6/hDNMT3B-202) with an average residue weight of 112.252 g/mol, charge of 25, isoelectric point of 8.4896, and molecular weight of 95,751.00 g/mol [56,60]. Recently, a total of 20 news transcripts were identified and annotated in Ensembl and HAVANA databases, including DNMT3B1, DNMT3B2, DNMT3B3, DNMT3B4, and DNMT3B5 [51,53,56,63,64]. Specifically, seven transcripts were annotated with the potential to code proteins, five transcripts with nonsense-mediated decay, two transcripts classified as protein-coding CDS not defined, and six transcripts considered as retained introns [56].

The *DNMT3L* gene is located at the 21q22.3 chromosome in humans, consists of 12 exons, transcribes an mRNA of 1706 bp (ENST00000270172.7/hDNMT3L-201) from the reverse strand (Figure 2E), and encodes a protein of 387 amino acids (43.583 kDa), with an average residue weight of 112.842 g/mol, charge of −4.0, isoelectric point of 5.6648, and molecular weight of 43,669.95 g/mol [52,56]. A total of four transcripts have been reported and annotated in Ensembl and HAVANA databases from this gene: three are classified as protein-coding and one as protein-coding CDS not defined [56].

At the protein level, the N-terminal region of the DNMT1 enzyme, also called the regulatory region, is composed of several domains, including the Charge-rich DNMT-Associated Protein (DMAP), Proliferating Cell Nuclear Antigen Binding (PBD), and Nuclear Location Sequence (NLS) domains. The DMAP domain is associated with the transcriptional repressor DNA Methyltransferase 1-Associated Protein 1 (DMAP1) and controls the stability and binding of DNMT1 to DNA. The PBD domain mediates the interaction with PCNA, leading to the association of DNMT1 with replication machinery. The NLS domain participates in the nuclear localization of DNMT1. The Replication Foci Targeting Sequence (RFTS) domain induces DNMT1 recruitment to the replication foci. The zinc finger (CXXC) domain participates in the allosteric regulation of DNMT1 activity, as well as the binding of DNMT1 to hemi-methylated DNA. Moreover, the DNMT1 enzyme contains a Poly Bromo Homology Domain (PBHD) composed of Bromo-Adjacent Homology 1 and 2 (BAH1/2) sub-domains that promote DNMT1′s interaction with additional proteins [43,65,66]. On the other hand, the DNMT2 enzyme lacks the N-terminal region, and DNMT3A and DNMT3B enzymes only have a Pro-Trp-Pro-Trp (PWWP), a cyst-rich domain, and an atrx-DNMT3-DNMT3L (ADD) domain, each in their N-terminal regions. Specifically, the PWWD domain participates in DNMT3A and DNMT3B interaction with chromatin [67,68,69], and the ADD domain regulates DNMT3A and DNMT3B activity [70,71]. Finally, the DNMT3L enzyme has an ADD and NLS domain in its N-terminal region [52,72] (Figure 3).

The C-terminal region of DNMT1, also named the catalytic region, participates in binding to a cofactor known as S-adenosyl-l-methionine (AdoMet or SAM), and this region is composed of ten conserved motifs, of which only six are conserved in mammals, and a Target Recognition Domain (TRD) located between VIII and IX motifs, which participate in the recognition of CpG to be methylated. Moreover, the IX motif promotes the binding of DNMT1 to DNA, while the IV and VI motifs participate in the nucleophilic attack on the cytosine ring to initiate methylation [66,73]. In contrast, DNMT2 contains only six conserved motifs (I, IV, VI, VIII, IX, and X), as well as a CFTXXYXXY motif (CFT) located between VIII and IX motifs [74]. Similarly, DNMT3A and DNMT3B enzymes contain the I, IV, VI, VIII, IX, and X conserved motifs [60], while the DNMT3L enzyme only contains I, IV, and VI motifs [52,72] (Figure 3).

## 4. Classification and Function of DNMT Enzymes

DNMT enzymes are classified into canonical and non-canonical types. Canonical DNMT enzymes have catalytic activity and are sub-classified as “maintenance” or “de novo”, while non-canonical enzymes lack catalytic activity. The DNMT1 enzyme is classified as a canonical and “maintenance” enzyme. This enzyme has a 10–40-fold preference for hemi-methylated DNA and converts it into fully methylated DNA during semi-conservative DNA replication with the help of two proteins: Proliferation Cell Nuclear Antigen (PCNA) and Ubiquitin-like with PHD and Ring Finger Domains 1 (UHRF1). Therefore, DNMT1 is important in the maintenance of DNA methylation [18,43,75]. DNMT2 is a non-canonical enzyme and does not participate in DNA methylation “de novo” or “maintenance”. However, it methylates cytosine 38 on the anticodon loop of aspartate in tRNA [18,76,77]. This modification protects tRNAs against stress-induced cleavage, preventing their fragmentation [78,79]. DNMT3A and DNMT3B are considered canonical and “de novo” DNMT enzymes [50]. On the other hand, the non-canonical enzyme DNMT3L lacks catalytic activity; however, it enhances the catalytic activity of DNMT3A and DNMT3B. Moreover, DNMT3L increases the binding of methyl groups to DNMT3A and DNMT3B and stabilizes them [18,76,77].

DNMT enzymes covalently transfer a methyl group from the S-Adenosyl Methionine (SAM) cofactor molecule to the C-5 position of the cytosine ring in CpG dinucleotide context to form 5-mC and S-Adenosyl-l-Homocysteine (AdoHcy or SAH) [77] (Figure 4).

DNMT enzymes participate in key cell processes and cancer, such as migration, invasion, apoptosis, and proliferation. Therefore, it is very important to understand the molecular mechanisms involved in the alterations in the expression and activity of these enzymes [57].

DNMT1 acts via the Dnmt1-PCNA-UHRF1 complex, and the consensus sequence for DNA methylation identified is ATTCGTCA (known as the untargeted mechanism). On the other hand, this enzyme can act via Dnmt1/transcription factor interactions (known as targets), including transcription factors such as SP1 and LSF. These mechanisms allow DNMT1 to cause aberrant DNA methylation and gene expression [80,81].

The DNMT3A enzyme methylates the (T/A/C)(A/T)(T/G/A)CG(T/G/C)G(G/C/A) consensus sequence via the transcription factor–DNMT3A–DNMT3L–DNA complex. Mechanistically, DNMTA interacts with oncogenic transcription factors, such as c-Myc, to specifically inhibit the expression of its target genes via methylation aberrations on their promoters (known as specific-site), including miRNAs and TSG, such as miR-200b and CDKN1A [82,83,84].

The DNMT3B enzyme specifically recognizes and methylates the (A/C)(C/G/A)(A/G)CGT(C/G)(A/G) consensus sequence through its physical interaction with the DNMT3L enzyme and oncogenic transcription factors, as SP1 and SP4, favoring transcriptional repression via targeted promoter hypermethylation [82,85].

## 5. Role of DNMT1 in CC

DNMT1 expression and activity increase in CC tissue samples and SiHa, HeLa, C-33A, and CaSki CC cell lines compared with cervical normal tissue samples and HcerEpci cells. Moreover, high DNMT1 expression correlates with poor Overall Survival (OS) and advanced stages in patients with CC [86,87,88,89,90,91,92,93,94,95,96]. Surprisingly, its expression is markedly elevated in HPV^+^ CC tissue samples compared to HPV^−^ CC tissue samples [88], suggesting that HPV could be involved in its deregulation in CC.

Various studies have shown that DNMT1 expression gradually increases from cervical normal tissue to LSIL, HSIL, and CC [88,97,98]. Therefore, DNMT1 expression could be useful as a diagnostic and prognostic biomarker in CC patients [88,96]. In addition, a novel method for DNMT1 protein detection in HeLa living cells was recently developed in vitro using an inhibitor of DNMT1 enzymes, known as RG108, and FITC, a bio-label with high biocompatibility and high quantum efficiency. Several RG108 probes conjugated with FITC were synthesized, and probe 8a showed that it is capable of sensing and monitoring the DNMT1 enzyme in the nucleus of HeLa living cells [99]; however, studies in a population are necessary to validate this novel method. On the contrary, the methylation level in the *DNMT1* promoter decreases gradually from the normal cervix to CC in tissue samples and correlates with plasma samples [88]. These findings suggest that aberrant DNMT1 expression may result from demethylation of its promoter in this type of cancer. Moreover, the methylation status of the *DNMT1* promoter could serve as a non-invasive diagnostic biomarker in cervical cancer.

It is well established that High-Risk HPV16 and HPV18 (HR-HPV16 and HR-HPV18) regulate gene expression through aberrant promoter methylation in cervical cancer cell lines such as SiHa, CaSki, and HeLa. This includes genes such as *CCND1* and the long non-coding RNA *MAGI2-AS* [100]. Mechanistically, a study showed that E6/E7 HPV16 overexpression increases DNMT1 expression in HaCaT cells [101]. On the other hand, another study reported that overexpression of high-risk HPV16 and HPV18 (HR-HPV16 and HR-HPV18) increases DNMT1 expression via the E7 oncoprotein in organotypic raft cultures of primary human foreskin keratinocyte (PHFK) cells. Furthermore, in PHFK cells, DNMT1 induces changes in DNA methylation patterns, including hypermethylation at genomic hotspots where HR-HPV integrates, regions of chromosomal loss, promoters with high CpG content, and bivalent genes marked by H3K4me3 and H3K27me3—such as *RARB* and *CADM1*. Interestingly, methylation of the *RARB* promoter mirrors the natural progression of cervical infection with HR-HPV16 and HPV18 in young women [102]. Another study reported that HR-HPV16, 18, and 31 overexpression increases DNMT1 expression in primary keratinocyte cells. Specifically, E6 and E7 overexpression of HR-HPV31 and 18 increases DNMT1 expression, and E7 overexpression of HR-HPV16 increases DNMT1 expression in primary keratinocytes through its interaction with ubiquitin ligase E6-Associated Protein (E6AP), thus preventing the degradation of DNMT1 via the proteasome. Surprisingly, DNMT1 knockdown decreases the expression of *E1* and *E4* viral genes, key genes to the viral life cycle; moreover, treatment with 5AZA-dC (a DNA methylation inhibitor) reduces clonogenic ability and the amount of papilloma virus DNA in CIN-612 cells [103], suggesting feedback in this cell system. In addition, HR-HPV16 E6 knockdown increases p53 expression, which decreases DNMT1 expression in SiHa and CaSki CC cells. However, DNMT1 re-expression promotes cell viability in SiHa CC cells [104,105]. Similarly, E6 and E7 knockdown decreases DNMT1, DNMT3A, DNMT3B and DNMT3L expression in SiHa CC cells, decreasing its viability and increasing apoptosis [106].

Interestingly, the interaction between DNMT1 and the HR-HPV19 E7 protein has been tested through the CR3 zinc finger domain of E7. This interaction stimulates DNMT1 activity in HeLa CC cells and in vitro [107]. On the other hand, the HR-HPV16 E7 protein forms a complex with DNMT1 to inhibit CCNA1 expression via methylation of its promoter in SiHa and C-33A CC cells [108]. HR-HPV16 E7 overexpression decreases C1QBP, BCAP31, CDKN2A, and PTMS expression and increases the methylation level in their promoters in HEK293 cells. At the mechanistic level, HR-HPV E7 interacts with the transcription factors GABPA, SP1, and ELK1, as well as with DNMT1, forming a multiprotein complex that targets and methylates specific genomic regions. This complex has been shown to methylate the promoters of *C1QBP*, *BCAP31*, *CDKN2A*, and *PTMS* in HEK293 cells. Comparable methylation events have also been identified in SiHa cervical cancer cells [109]. HPV31b, another high-risk HPV type, has been shown to increase DNMT1 expression in primary keratinocytes upon overexpression. However, DNMT1 expression decreases following cellular differentiation. A similar decrease in DNMT1 levels upon differentiation has also been observed in CIN-612 cells [103].

A study revealed that DNMT1 knockdown increases the proportion of cells in the G0/G1 phase, decreases the proportion in the S phase, promotes apoptosis, and reduces cell proliferation in HeLa and SiHa cervical cancer cells. Mechanistically, DNMT1 knockdown leads to hypomethylation and upregulation of TSG (*CCNA1*, *PTEN*, *CHFR*, *PAX1*, *SFRP4*, and *TSLC1*) in both cell lines [110]. Similarly, DNMT1 knockdown increases Cep131 expression via demethylation of its promoter, which enhances the recruitment of SP1 transcription factor to the *Cep131* promoter in HeLa CC cells. Subsequently, Cep131 overexpression inhibits cell viability and the growth of tumors in vivo and in vitro [111]. Interestingly, the transcription factor SP1 upregulates DNMT1 expression, leading to the repression of *EphA7* through aberrant promoter methylation. Re-expression of *EphA7* has been shown to reduce cell proliferation, migration, and Epithelial–Mesenchymal Transition (EMT) while promoting apoptosis in CaSki and SiHa cervical cancer cell lines [112]. On the other hand, knockdown of DNMT1 and DNMT3B reduces methylation of the *SORBS3-β* promoter, thereby increasing its expression in SiHa cervical cancer cells. Subsequently, *SORBS3-β* re-expression inactivates the Wnt/β-catenin signaling pathway through UBA-mediated ubiquitination of β-catenin, ultimately suppressing lymphatic metastasis of cervical cancer both in vivo and in vitro [113]. Finally, DNMT1 knockdown increases E-cadherin expression and decreases methylation in its promoter in SiHa, HeLa, and C-33A CC cell lines [86]. Consistently, previous studies have shown that E-cadherin expression decreases in plasma and tissue samples of patients with CC due to aberrant methylation [86,114].

## 6. Role of DNMT2 in CC

Currently, very little is known about the DNMT2 enzyme in CC, including its expression level, regulation mechanisms, and the biological processes regulated by this enzyme. Only two studies have evaluated the role of the DNMT2 enzyme in CC [115,116]. The first reported that *DNMT2/TRDMT1* gene knockout promotes chemotherapy-induced senescence via Doxorubicin (DOX) and Etoposide (ETOPO) in HeLa CC cells, which increases Azacytidine-mediated, apoptosis-based senolytic activity and increases 53BP1 and p62 expression, as well as the synthesis of lysosomes. Additionally, ETOPO induces cell cycle arrest at the G0/G1 phase and increases IL-6 levels while simultaneously reducing DNA damage, micronuclei formation, LC3 expression, and cytoplasmic 5-mC levels. DOX increases apoptosis; the production of ROS; and the expression of RAD51, XRCC1, NF-κB, and cytoplasmic 5-mC levels. On the other hand, this drug decreases NSUN1 levels [115]. Later, a second study showed that DOX treatment decreases metabolic activity, caspase-3 activation, and proteasomal activity while increasing oxidative protein damage; protein aggregates; GRP78, NSUN3, and NSUN5 levels; and AMPK activation in HeLa CC cells with *DNMT2/TRDMT1* gene knockout [116]. Therefore, these studies reported the potential use of the DNMT2 enzyme as a therapeutic target in CC.

Accordingly, it is crucial to determine whether DNMT2 expression is altered in cervical cancer; the potential impact of high-risk HPV infection on this enzyme; and the involvement of other molecules, such as transcription factors, miRNAs, long non-coding RNAs (lncRNAs), or circular RNAs (circRNAs). Furthermore, elucidating the biological and molecular processes regulated by DNMT2, particularly in this cancer type, will be of great interest.

## 7. Role of DNMT3A in CC

DNMT3A expression is increased in CC tissue samples compared with normal cervical tissue samples [91,117]. Interestingly, DNMT3A expression is higher in C-33A CC cells compared to HeLa and CaSki CC cells [118] and CaSki CC cells compared with HeLa CC cells [119]. DNMT3A expression decreases in CC patients with integrated HR-HPV16 compared to HPV-negative controls. Surprisingly, DNMT3A expression increases in CC patients with the methylated immunostimulatory motif of E6 and E7 regions and low hypomethylation within Alu repeat sequences, and its expression decreases in CC patients with the same unmethylated motif and high DNA hypomethylation at the Alu repeat sequences compared with HPV-negative controls, suggesting that DNMT3A could favor a differential molecular signature between patients with CC and infection by HR-HPV16 [120].

Notably, the *DNMT3A* promoter is not methylated in CC tissue or normal samples [121], suggesting that the regulation of DNMT3A expression is not regulated by methylation. In contrast, a study reported that depletion of the methyl donors, such as folate and methionine, decreases DNMT3A and DNMT3B expression, as well as the percentage of global DNA methylation in C-4 II and SiHa CC cells. However, methyl donor repletion reverts these changes [122].

DNMT3A regulates gene expression by directly binding to promoters, such as the *CCNA1* promoter in C-33A CC cells [108]. Similarly, CUL4B recruits the SUV39H1/HP1/DNMT3A complex to the *IGFBP3* promoter, inhibiting its expression via aberrant methylation. This process promotes cell proliferation and invasion while inhibiting apoptosis in HeLa, SiHa, and CaSki CC cells and additionally supports in vivo tumor growth. Consistently, the expression of CUL4B at both the mRNA and protein levels is increased in tissue samples from CC patients, and its expression negatively correlates with IGFBP3 expression [123]. In line with these findings, DNMT3A binds to the *TP53AIP1* promoter, increasing its methylation level and reducing its expression in SiHa and HeLa CC cell lines. Moreover, DNMT3A knockdown induces the re-expression of TP53AIP1, leading to decreased metastasis and reduced in vivo tumor growth [117].

In contrast, E6 and E7 of HR-HPV18 increase EZH2 expression, which is the enzymatic subunit of PRC2 complex and catalyzes the tri-methylation of lysine 27 on histone H3 (H3K27me3), an epigenetic mark associated with transcriptional repression [124]. EZH2 downregulates DNMT3A expression through the H3K27me3 transcriptional repression epigenetic mark in HeLa CC cells. Subsequently, low DNMT3A expression decreases methylation levels in *HAVCR2* and *LGALS9* promoters, promoting Tim-3 and Galectin-9 expression in HeLa and C-33A CC cells [125], suppressing the anti-tumor immunity mediated by innate and adaptive immune cells [126]. Similarly, EZH2 knockdown decreases H3K27me3 levels in the *DNMT3A* promoter, increasing its expression. Subsequently, DNMT3A binds *CCLL2* and *CCR4* promoters and inhibits their expression by aberrant methylation, inhibiting cell migration and EMT in vitro and in vivo [127]. On the other hand, SUV39H1 binds to the *DNMT3A* promoter and increases the H3K9me3 transcriptional activation epigenetic mark, increasing DNMT3A expression in SiHa and HeLa CC cells. Then, DNMT3A binds to promoters of *HAVCR2* and *LGALS9* genes, increasing the methylation levels in these promoters and decreasing the expression of Tim-3 and Galectin-9 in SiHa and HeLa CC cells, as well as in vivo tumor growth. Surprisingly, E6 and E7 of HR-HPV16 and HR-HPV18 do not affect the SUV39H1 expression or H3K9me3 levels in C-33A, HeLa, and SiHa CC cells [128].

DNMT3A knockdown increases BST2 expression via demethylation of its promoter in SiHa and HeLa CC cell lines. BTS2 promotes CC in vivo and in vitro [129].

## 8. Role of DNMT3B in CC

DNMT3B expression is increased in CC tissue samples and C-33A, HeLa, SiHa, and CaSki CC cells compared with cervical normal tissue samples and HaCaT cells [91,119,130]. DNMT3B expression is increased in HR-HPV16-positive CC patients compared with HPV-negative controls, and its high expression is more marked in CC patients with episomal HR-HPV16 compared with CC patients with integrated HR-HPV16. Interestingly, DNMT3B expression increases in CC patients with the methylated E6 motif compared to HPV-negative controls [120].

A study showed that the *DNMT3B* promoter is not methylated in CC tissue or normal samples [121], suggesting that its expression is not altered by aberrant methylation. In contrast, Single-Nucleotide Polymorphisms (SNPs) in *DNMT3B* promoters alter their expression in cancer. Particularly, the CT genotype of the 46359CT polymorphism is associated with CC risk. This SNP is associated with a new binding site to the GATA-1 oncogenic transcription factor, which increases DNMT3B expression. Moreover, the TT genotype is associated with a decreased risk of HSIL and LSIL in the Mexican population [131,132]. On the other hand, the copy number of the *DNMT3B* gene is increased in CC patients from Amsterdam and SiHa CC cells, which correlates with an increase in its expression [133].

FOXO3a is a transcription factor, and its overexpression decreases cell proliferation, migration, and invasion and promotes apoptosis in HeLa CC cells. Specifically, FOXO3a decreases DNMT3B expression through its direct binding to FOXO3a-E binding element (consensus DNA sequence [G/C/A] [T/C/A] AAA [T/C] A) present in the *DNMT3B* promoter, leading to the re-expression of *PTEN* TSG via demethylation of its promoter in HeLa CC cells [134]. Similarly, DNMT3B knockdown decreases cell proliferation, migration, and invasion in HeLa3rd (a HeLa subline) and CaSki, as well as in vivo tumor growth and metastasis. Mechanistically, DNMT3B knockdown decreases its binding to the *PTPRR* promoter, increasing the PTPRR expression via demethylation of its promoter in HeLa3rd and CaSki CC cells. PTPRR overexpression inhibits the MAPK signaling pathway; the expression of E6 and E7 oncogenes of HR-HPV16 and HR-HPV18, DNMT1, DNMT3A, and DNMT3B; as well as EMT genes in HeLa3rd and CaSki CC cells. Moreover, PTPRR overexpression decreases in vivo tumor growth and metastasis [119]. However, E7-HR-HPV16 overexpression does not affect DNMT3B expression, which binds to the *CCNA1* promoter in the C-33A CC line [108]. In contrast, another study reported that DNMT3B expression increases in the C-33A CC cell line with E6 and E7-HR-HPV16/18 overexpression via a recombinant adenovirus system, suggesting that high DNMT3B expression is associated with HR-HPV infection and could favor CC development [135]. However, these results contradict the E7-HR-HPV16 effect in DNMT3B expression, but this could be explained by the system of expression used in each study (transient expression—pcDNA3.1/myc-HIS vs. stable expression—pBR322-Ad4-E1Amut adenovirus plasmids).

Finally, DNMT3B knockdown decreases methylation in the *BST2* promoter and increases its expression in SiHa and HeLa CC cell lines. Furthermore, BTS2 promotes viability and EMT in these cervical cancer cell lines, as well as tumor growth in vivo, while simultaneously inhibiting apoptosis in vitro [135].

## 9. Role of DNMT3L in CC

Currently, little is known about the role of the DNMT3L enzyme in CC, including its expression. Likewise, a study reported that the methylation level in the *DNMT3L* promoter decreases in CC tissue samples compared to normal tissue samples. Interestingly, methylation in the *DNMT3L* promoter is minor in HeLa CC cells compared with SiHa CC cells, and this inversely correlates with its expression in these CC cells [121], suggesting that aberrant methylation on the *DNMT3L* promoter could be associated with its altered expression in CC.

On the other hand, Trichostatin A (TSA) is a hydroxamic acid produced by *S. hygroscopicus* that acts as a reversible inhibitor of Histone Deacetylases (HDAC) 1, 2, 3, 4, 6, 7, and 9 through the sequestering of zinc ions at the active site of HDACs [136,137]. Interestingly, TSA and 5-Azacytidine increase DNMT3L expression via demethylation of its promoter in HeLa and SiHa CC cells. Subsequently, DNMT3L overexpression increases cell proliferation and anchorage-independent growth, promotes a change in phenotype, and supports an increase in nuclear reprogramming (which includes overexpression of DNMT1 and DNMT3B in HeLa CC cells) [138]. However, the downstream effects are completely unknown.

## 10. DNMTs as Therapeutic Target Potential in CC

Currently, several studies have investigated the role of DNMTs as therapeutic targets in this type of cancer. The drugs evaluated regulate DNMT enzymes by decreasing their enzymatic activity (such as Betulin), expression (such as Zeb), or both (such as EGCG) in CC (Table 1). Moreover, these drugs are synthetic compounds, such as 5-Aza-dC and CDDP, or natural compounds, including Luteolin and EGCG (Table 1).

According to the literature, the drugs more commonly studied are inhibitors of DNMT enzymes, such as 5-Aza-dC; however, despite the great effort and encouraging results, there is still a significant amount of work to be completed. For example, several research groups reported contradictory results regarding the role of 5-Aza-dC in CC [88,139,140,141]. On the one hand, it was reported that 5-Aza-dC decreases the expression and activity of DNMT1, DNMT3A, and DNMT3B enzymes in HeLa and C-33A CC cells [139,140,141]. Conversely, another study showed that the 5-Aza-dc does not affect DNMT1 expression in HeLa and SiHa CC cells [88]. These results could be explained by factors such as the frequency of cell passage [142], the culture medium used, and treatment time with this drug. Below, we present the studies conducted on the drugs evaluated as potential treatments in CC (Table 1).

Thus far, many compounds have been proven to suppress the expression or enzymatic function of DNMT enzymes, which could be good candidates for future treatments for CC. Nevertheless, the effectiveness of these drugs still needs to be evaluated in more CC cell lines, as well as in mouse models (in vivo). Similarly, many disadvantages need to be addressed, such as the off-target with other enzymes or cytotoxicity impact in non-malignant cells, to ensure its effectiveness and safety.

**Table 1 ijms-26-10496-t001:** Drugs targeting DNMT enzymes as potential treatments in CC.

Drug	Description	Results	Model	Cite
5-Aza-2′ deoxycytidine (5-Aza-dC)	Nucleoside analog and DNA methylation inhibitor	Decreases the expression and activity of DNMT1, DNMT3A, and DNMT3B	HeLa and C-33A CC cell lines	[139,140,141]
5-Aza-dC	Nucleoside analog and DNA methylation inhibitor	Does not affect DNMT1 expression	HeLa and SiHa CC cell lines	[88]
Zebularine (Zeb)	Nucleoside analog of cytidine and DNA methylation inhibitor	Decreases DNMT1, DNMT3A, and DNMT3B expression, but unfortunately, this drug does not decrease their activity	HeLa CC cell line	[143]
Trichosanthin (TCS)	Type I Ribosome-Inactivating Protein that is extracted from the Trichosanthes kirilowii herb	Inhibits the expression and activity of DNMT1, reducing methylation and increasing the expression of APC and TSLC1 TSG	HeLa and CaSki CC cell lines	[144]
Apicidin (N-O-methyl-L-tryptophanyl-L-isoleucinyl-D-pipecolinyl-L-2-amino-8-oxodecanoyl)	Anti-protozoal fungal metabolite agent that acts as a selective HDACs inhibitor for Class I HDACs (HDACs 2, 3, and 8)	Reduces DNMT1 expression by decreasing the levels of acetyl-Histone 3 and acetyl-Histone 4; decreasing the levels of transcriptional activation markers, such as H3K4me3; and increasing the levels of H3K9me3 and H3K27me3, two marks of transcriptional repression. It promotes pRB and HDAC1 recruitment and avoids the binding of Pol II and P/CAF (or KAT2B), a Histone Acetyltransferase (HAT) enzyme that adds acetyl groups onto lysine residues in histone proteins. Subsequently, DNMT1 downregulation increases p21 expression and induces apoptosis	HeLa CC cell line	[145,146]
Luteolin (3′,4′,5,7-tetrahydroxyflavone) and *Limoniastrum guyonianum*	Luteolin is a dietary flavonoid, and *L. guyonianum* is a plant; specifically, it is a shrub from the *plumbaginaceae* family	Decrease UHRF1 and DNMT1 expression and global DNA methylation while increasing p16 expression, inhibiting cell proliferation, and inducing arrest of the cell cycle in the G2/M phase and apoptosis	HeLa CC cell line	[147]
Epigallocatechin gallate (EGCG), Eugenol, and Amarogentin	Major component of green tea polyphenols; active component of clave and chirata plant	These 3 drugs decrease DNMT1 and increase RBSP3, LIMD1, and p16 protein expression via demethylation in *RBSP3* and *p16* promoters, which inhibit cell proliferation, block the cell cycle in the G1/S phase, and increase apoptosis	HeLa CC cell line	[148]
EGCG	Major component of green tea polyphenols	Decreases the expression and activity of DNMT1, DNMT3A, and DNMT3B enzymes through competitive binding to the catalytic cavity of DNMT1 and DNMT3A, decreasing global DNA methylation and increasing the expression of *RARβ*, *CDH1*, and *DAPK1* TSG via demethylation of their promoters	HeLa CC cell line	[141,149]
Menthol	Cyclic monoterpene	Decreases DNMT1 activity and increases FANCF expression via demethylation in its promoter, contributing to the maintenance of genome integrity	SiHa CC cell line	[150]
Folate (F) and Methionine (M)	Vitamin B9 and indispensable dietary amino acid	Their depletion reduces cell proliferation by decreasing DNMT1, DNMT3A, and DNMT3B expression, as well as global DNA methylation	SiHa and C-4II CC cell lines	[122]
H1	2-(2-aminobenzo[d]thiazol-6-yl) benzo[d]oxazol-5-amine derivative	Inhibits cell proliferation and tumor growth and promotes cell cycle arrest via repression of the E7/Rb/E2F-1/DNMT1 signaling pathway	HeLa CC cell line and male BALB/c nude mice	[151]
Capsaicin	Capsaicinoid present in chili peppers	Inhibits DNMT1 activity and methylation on promoters of *CADM1* and *SOCS1* genes while increasing cell viability and the formation of apoptotic bodies	HeLa CC cell line	[152]
Casticin (3′,5-dihydroxy-3,4′,6,7-tetramethoxyflavone or CAS)	7-O-methylated flavonoid extracted from Fructus viticis	Represses DNMT1 activity and expression and reverts the stemness phenotype	Cancer stem-like cells derived from the HeLa and CaSki CC cell line	[93]
Ethanol	Ethyl alcohol	Decreases acid folic levels and increases DNMT1, DNMT3A, and DNMT3B expression, leading to in vivo genome-wide hypomethylation	SiHa CC cell line and athymic nude mice	[153]
Laminarin	Polysaccharide found in brown algae	DNMT1 could be a target of Laminarin in CC and COVID-19 patients	In silico	[154]
Quercetin	Dietary phytochemical	Interacts with DNMT3A and DNMT3B and inhibits their activity. It also decreases DNMT1, DNMT3A, and DNMT3B expressions, as well as global DNA methylation, leading to the re-expression of TSGs, such as *RARB*, *TIMP3*, *VHL*, *PTEN*, *CDH1*, and *SOCS1* via demethylation of their promoters	HeLa CC cell line	[155]
Valproic acid (VPA)	Inhibitor of HDAC1	Treatment inhibits the formation of the DNMT3A/HDAC1 complex, leading to OCT4 re-expression	C-33A CC cell line	[156]
Sulforaphane (SFN)	Dietary phytochemical	Inhibits DNMT activity and decreases DNMT3B expression, which induces the expression of *RARβ*, *CDH1*, *DAPK1,* and *GSTP1* TSG via demethylation of their promoters	HeLa CC cell line	[140]
Betulin	Triterpenoid	Inhibits DNMT3A activity and reduces cell viability	HeLa CC cell line	[157]
Cisplatin (Cis-diaminodichloroplatinum (II) or CDDP)	Alkylating agent	Decreases DNMT1 expression, promoting the re-expression of BRCA1, BRCA2, FANCC, and FANCD2 genes via demethylation of their promoters	HeLa and SiHa CC cell lines	[89]
CDDP	Alkylating agent	Inhibits DNMT1 expression, increases BRCA1, BRCA2, FANCC, and FANCD2 expression, and decreases the methylation levels in their promoters	SiHa and HeLa CC cell lines	[89]
CDDP	Alkylating agent	DNMT3B expression increases in CDDP-resistant subclones. However, co-treatment with CDDP and 5-Aza-CdR only increases DAP kinase expression to the mRNA level via demethylation of its promoter in these CDDP-resistant subclones	ME180 CC cell line	[158]
SGI-1027 (N-(4-(2-Amino-6-methylpyrimidin-4-ylamino)phenyl)-4-(quinolin-4-ylamino) benzamide)	Quinoline-based compound	Inhibits DNMT1 activity, increasing apoptotic cell death and cell cycle arrest, and decreasing viability in vitro and in vivo by inhibition of the JAK/STAT signaling pathway	HeLa CC cell line	[159]
Trichostatin A (TSA)	HDAC inhibitor	Inhibits DNMT3B expression and induces apoptosis	CaSki and HeLa CC cell lines	[160]
Hydralazine	Antihypertensive	Induces APC expression and promotes demethylation, inhibiting cell growth	HeLa, CaSki, and SiHa CC cell lines	[161]
Hydralazine	Antihypertensive	Increased the PFS in patients with advanced CC	19 Mexican patients	[162]
Hydralazine	Antihypertensive	Reverts gemcitabine resistance via inhibition of G9A histone methyltransferase in vitro	CaLo CC cell line	[163]
Genistein	Polyphenol	Inhibits the expression and enzymatic activity of DNMTs, reverts the promoter region methylation of the TSGs, and re-establishes their expression	HeLa CC cell line	[164]
Hydralazine	Antihypertensive	Induces radiosensitization and decreases cell viability	SiHa CC cell line	[165]
Hydralazine	Antihypertensive	Demethylates and reactivates the expression of TSG without affecting global DNA methylation in vivo	Patients with untreated CC	[166]

PFS: Progression-Free Survival.

## 11. The Interplay Between DNMTs, miRNAs, and lncRNAs in Cervical Cancer

It is well known that there is an interplay between the DNMT1 enzyme and microRNAs (miRNAs) [167]. In this context, DNMT1, DNMT3A, and DNMT3B enzymes bind to the *miR-142-5p* promoter and decrease its expression via methylation. In contrast, *miR-142-5p* overexpression inhibits proliferation, migration, invasion, and metastasis and promotes apoptosis through the PIK3AP1/PI3K/AKT signaling pathway in HeLa and C-33A CC cells [139]. Moreover, DNMT1 knockdown increases *miR-484* expression via demethylation of its promoter, decreasing cell migration, invasion, EMT, cell adhesion, and tumor growth via downregulation of β-integrin, WNT/MAPK, and TNF-α signaling pathways in HeLa and C-33A CC cells [168]. Also, *miR-148b* overexpression decreases DNMT1 expression in HeLa CC cells [169]. Similarly, *miR-29a* overexpression decreases DNMT1 expression through direct binding to DNMT1 3′-UTR, which increases SOCS1 TSG expression via demethylation of its promoter. SOCS1 inhibits the NF-κB signaling pathway, decreasing cell proliferation, migration, invasion, and EMT and promoting apoptosis in SiHa and HeLa CC cells [90]. Interestingly, HR-HPV16 E6 knockdown increases *miR-23b* expression via demethylation of its host gene C9orf3 by DNMT1 downregulation in SiHa cells. In contrast, *miR-23b* re-expression inhibits the c-Met/Akt signaling pathway and induces apoptosis in SiHa CC cells [170]. *miR-148a-3p* targets 3′-UTR of DNMT1 and decreases its expression in HeLa and SiHa cells. Moreover, DNMT1 knockdown induces UTF1 expression via demethylation of its promoter in HeLa and SiHa CC cells [87]. Finally, DNMT1 plays a key role in maintaining cervical cancer stem cell-like cells (CCSLCs) through the *miR-342-3p*/DNMT1/FoxM1 regulatory axis. In HeLa-derived CCSLCs, *miR-342-3p* expression is downregulated by methylation of its promoter. However, overexpression of *miR-342-3p* reduces the expression of its targets, DNMT1 and *FoxM1*, thereby inhibiting self-renewal-associated stemness phenotypes in vitro and suppressing tumor growth. Interestingly, DNMT1 knockdown leads to increased *miR-342-3p* expression, suggesting the presence of a feedback loop between DNMT1 and *miR-342-3p* [171].

Like DNMT1, miRNAs also regulate DNMT3A expression in CC. For instance, overexpression of *miR-182* inhibits DNMT3A expression and promotes apoptosis in C4-II cervical cancer cells; however, re-expression of DNMT3A reverses this effect [172]. In a similar manner, *miR-29a* downregulates DNMT3A and DNMT3B expression in HeLa and C-33A cervical cancer cells, which results in the demethylation and subsequent upregulation of p16. This upregulation contributes to the inhibition of cell proliferation and the induction of cell cycle arrest [173]. *miR-331-3p* overexpression through Bone Marrow Mesenchymal Stem Cell-derived Extracellular Vesicles inhibits DNMT3A expression and promotes LIMS2 re-expression via demethylation of its promoter in CaSki and HcerEpic CC cells, which decreases cell proliferation, migration, and invasion and increases apoptosis [174].

Recently, it was proposed that DNA methylation is mediated by lncRNAs in cancer, given that these molecules recruit DNMTs to specific genomic sites or regions, such as gene promoters [175]. In this sense, LINP1 is an lncRNA that promotes cell proliferation and viability and decreases apoptosis by inhibiting the expression of *KLF2* and *PRSS8* genes via DNMT1. However, DNMT1 knockdown partially reverses these results in CaSki and C-33A CC cells [176]. SIX1-1 is another lncRNA that interacts with the DNMT1 enzyme and facilitates its binding to the *RASD1* promoter, inhibiting its expression via aberrant methylation. Subsequently, this cell proliferates in CaSki and ME180 CC cell lines, and in vivo tumor growth increases due to activation of the cAMP/PKA/CREB pathway [177]. HOTAIR lncRNA promotes the transcriptional activity of Wnt/β-catenin and PI3K/AKT oncogenic signaling pathways in the HeLa CC cell line. Mechanistically, this lncRNA interacts with the DNMT1 enzyme and recruits it to the *PTEN* promoter, a negative regulator of these two pathways, decreasing its expression via aberrant methylation [178].

On the other hand, knockdown of HOTAIR decreases DNMT3B expression, which, in turn, promotes *LATS1* expression via demethylation of its promoter. Subsequently, *LATS1* reduces nuclear accumulation of YAP1, inhibiting migration and invasion of HeLa cervical cancer cells, as well as tumor growth in vivo [179].

These studies showed that either directly or indirectly, DNMT enzymes inhibit the expression of tumor-suppressor lncRNAs and miRNAs and promote the expression of oncogenic lncRNAs and miRNAs in CC. At the same time, various tumor-suppressor lncRNAs and miRNAs inhibit DNMT expression, and several oncogenic lncRNAs and miRNAs promote its expression in CC, indicating a very complex molecular network.

In addition to the role of non-coding RNAs (ncRNAs) in promoting cervical cancer, recent evidence has shown that post-translational modifications of histones—such as acetylation and deacetylation—also contribute to the development of this tumor. A recent review addressed this topic in detail [36]. For example, the expression of CSRP2BP HAT enzyme increases in patients with CC and CC cell lines. Interestingly, CSRP2BP knockdown decreases proliferation and metastasis in vivo and in vitro. Mechanistically, CSRP2BP regulates N-cadherin transcription by recruiting SMAD4 in HeLa and C-33A CC cell lines [180].

## 12. DNA Methylation as Biomarker in CC

Several studies have shown that DNA methylation could be useful as a diagnostic and prognostic biomarker in CC. For example, a study revealed that methylation in the *EDN3* promoter decreases its expression in CC. However, treatment with 5-Aza-dC promotes its re-expression. Consistently, EDN3 overexpression inhibits proliferation, migration, and invasion in C-33A, SiHa, and CaSki CC cell lines [94]. On the other hand, methylation in the *EphA7* promoter gradually increases in normal, CINII/III, and CC tissue samples and could be a potential diagnostic biomarker in this type of cancer [181]. In addition, a high methylation level in the *EphA7* promoter is a potential immunotherapy biomarker [129], and methylation in *BRCA1*, *BRCA2*, *FANCC*, and *FANCD2* promoters was detected in plasma samples of patients with CC. Interestingly, methylation levels were most prevalent in advanced stages compared to early stages, and they correlate with their levels in primary tumors. Moreover, methylation in *BRCA1*, *BRCA2*, *FANCC*, and *FANCD2* promoters in patients with CC (advanced stages) correlated with poor survival, even after therapeutic intervention with cisplatin-based concurrent chemoradiotherapy, strongly suggesting that methylation in *BRCA1*, *BRCA2*, *FANCC*, and *FANCD2* promoters could be useful as a prognostic biomarker in CC [89].

Currently, a set of studies supports the use of DNA methylation in screening programs for CC. Numerous studies have elucidated the impact of this epigenetic modification on the diagnosis of patients with CC in 2025 (Table 2). Interestingly, the GynTect^®^ test was the most frequently used assay this year to diagnose women with CC. However, there are variations in the results, even though the study population is from the same country [182,183], which could be explained in part by the sample size and type of sampling (Table 2). On the other hand, methylation of the *ZNF671* gene promoter has been detected in several studies using different assays, such as GynTect^®^ and qMSP [182,183,184], strongly suggesting that it has potential as a diagnostic biomarker in CC (Table 2). Surprisingly, a study revealed an AUC of 0.9421 for methylation in the *ZSCAN18* promoter with three different assays and a total of 763 samples used for identification and validation cohorts [185], strongly suggesting that methylation in the promoters of these genes could be useful as a diagnostic biomarker in patients with CC (Table 2). Finally, another study reported aberrant methylation in FAM19A4 and *miR-124-2* in 100% of CC cases. However, the sample size was small [186] (Table 2).

In addition, in 2025, some studies demonstrated the potential use of methylation biomarkers for the prognosis of CC patients, employing techniques such as GynTect^®^ Arrays and qMSP (Table 3). However, all these studies need to be further validated in independent cohorts with other techniques and larger cohorts.

## 13. Future Directions

Future studies are necessary to elucidate the molecular mechanisms involved in the overexpression of DNMT enzymes in CC at multiple regulatory levels.

At the transcriptional level, potential mechanisms may include the involvement of molecules such as oncogenic transcription factors and lncRNAs, as well as copy number alterations in *DNMT* genes—mechanisms that have already been identified in other diseases [198]. Moreover, since DNMT gene promoters contain CpG islands, it is plausible that these enzymes could self-regulate via promoter methylation in CC, as observed in the aberrant methylation of the *DNMT1* promoter in glioblastoma [199].

At the post-transcriptional level, DNMT overexpression could be driven by the downregulation of several tumor-suppressive miRNAs and circRNAs, as has been reported in other biological contexts [198].

Additionally, the altered expression of DNMT isoforms plays a critical role in various human cancers due to their differing catalytic activities [200,201]. Therefore, it is essential to investigate the expression patterns of all DNMT isoforms in CC and assess their individual contributions to cervical carcinogenesis. This represents an important opportunity to further elucidate the molecular mechanisms regulating DNMT expression in CC.

It is also well established that DNMTs physically interact with various molecules, including non-coding RNAs such as lncRNAs [175,202]. For example, DNMT1 interacts with the lncRNA DACOR1 to regulate gene expression and DNA methylation in colon cancer [203]. Accordingly, understanding the mechanisms of DNMT recruitment to specific DNA-binding sites in the context of CC is of significant interest.

In addition to their catalytic roles, DNMT enzymes have been shown to possess non-catalytic functions in contexts outside of CC, including hematopoietic stem and somatic cells [200,204,205]. Thus, it is also important to investigate their catalytic-independent functions specifically in CC.

Furthermore, DNMT expression has been proposed as a therapeutic, diagnostic, and prognostic biomarker in several cancers, including non-small cell lung cancer [206]. Consequently, future research should focus on validating the potential use of elevated DNMT expression as a non-invasive biomarker in blood samples (serum or plasma) from CC patients, ideally using large, well-characterized patient cohorts.

Bioinformatic analyses have also revealed that DNMT expression correlates with key genes involved in CC. For instance, a negative correlation has been observed with PTEN expression [178], and a positive correlation has been observed with TYMS expression in CC patients [207]. However, the molecular implications of these associations require further investigation.

Finally, a recent study demonstrated that the catalytic domain of the TET1 enzyme can be used as a molecular tool to demethylate the BRCA1 gene promoter in MCF-7 cells [208]. This opens the door to exploring whether DNMT enzymes—or their catalytic domains—could be used as targeted molecular tools to repress the expression of oncogenes or other key genes implicated in cervical carcinogenesis.

## 14. Conclusions

Research on DNA methylation in CC is rapidly expanding and has identified DNMTs as key enzymes involved in aberrant DNA methylation. Their overexpression leads to the silencing of TSGs via promoter hypermethylation, thereby contributing to cervical tumorigenesis.

Furthermore, DNMT overexpression shows promise as an early diagnostic and prognostic biomarker in blood samples (serum or plasma) from patients with CC. Importantly, these enzymes may serve as promising therapeutic targets in CC treatment. However, strategies to minimize toxicity and maximize therapeutic efficacy must be further developed.

Although significant progress has been made, our understanding of the role of DNMTs in CC remains incomplete. Continued research in this area presents both challenges and significant opportunities for future investigation.

## Figures and Tables

**Figure 1 ijms-26-10496-f001:**
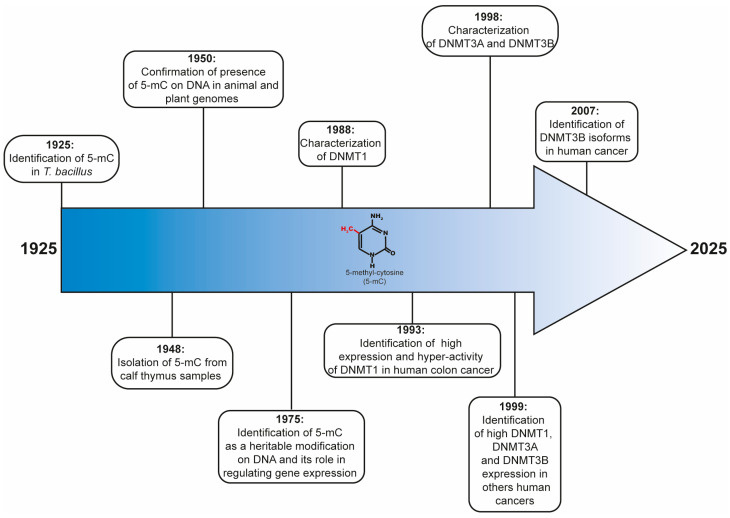
A timeline of the main findings related to DNMT enzymes and 5-mC. Methylated cytosines are shown in red.

**Figure 2 ijms-26-10496-f002:**
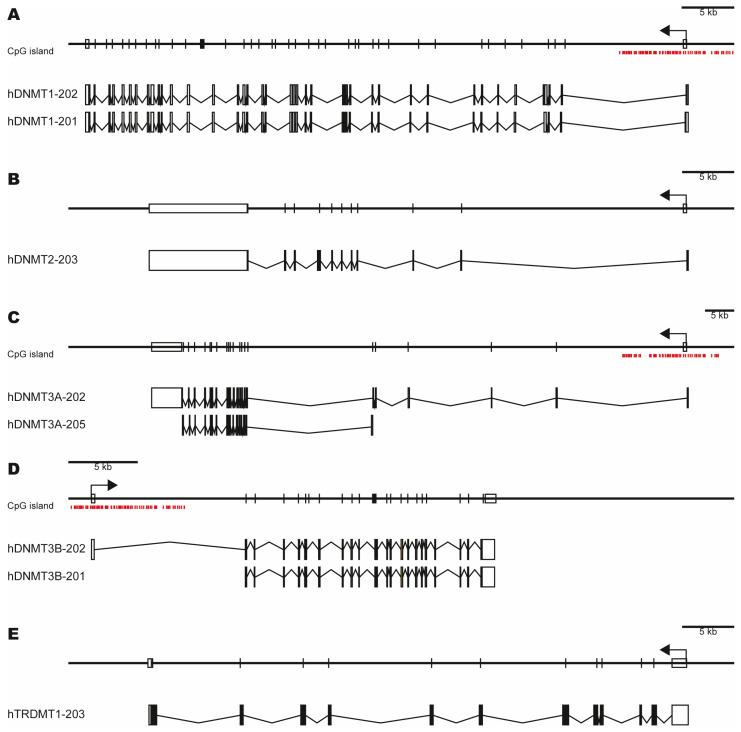
The structure of *DNMT* genes and their most abundant mRNAs. (**A**) The *DNMT1* gene is on the 19p13.2 chromosome and contains a CpG island and 41 exons. mRNA of DNMT1 isoform A contains 41 exons (ENST00000359526.9/hDNMT-202), and mRNA of DNMT1 isoform B contains 40 exons (ENST00000340748.8/hDNMT-201). (**B**) The *DNMT2* gene is located on the 10p12-10p14 chromosome and contains 11 exons (ENST00000377799.8/hTRDMT1-203/hDNMT2-203). (**C**) The *DNMT3A* gene is located on 2.p23 human chromosome and contains a CpG island and 23 exons. The ENST00000321117.10/hDNMT3A-202 transcript contains 23 exons, while the ENST00000402667.1/hDNMT3A-205 transcript contains only 18 exons. (**D**) The *DNMT3B* gene is located on 20q11.2 and contains a CpG island and 23 exons. The ENST00000328111.6/DNMT3B-202 transcript contains 23 exons, while the ENST00000201963.3/DNMT3B-201 transcript contains 22 exons. (**E**) The *DNMT3L* gene is on the 21q22.3 chromosome and contains 12 exons. Red lines: CpG island. Rectangle: exon. Arrow: transcription start site and orientation of transcription.

**Figure 3 ijms-26-10496-f003:**
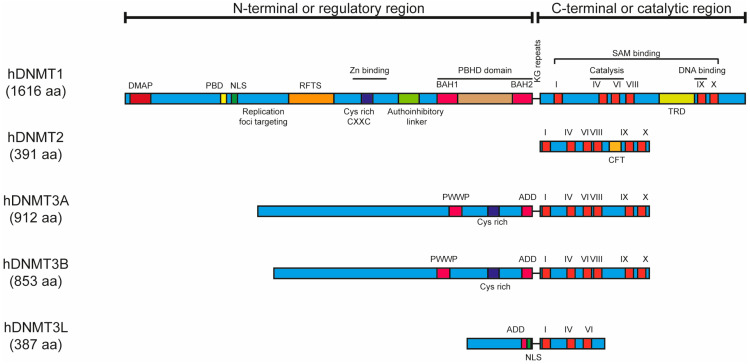
The structure of human DNMT enzymes. The regions and domains of the human DNMT enzymes are shown. DMAP: Charge-rich DNMT-Associated Protein domain. PBD: Proliferating Cell Nuclear Antigen (PCNA) Binding Domain. NLS: Nuclear Location Sequence. RFTS: Replication Foci Targeting Sequence. CXXC: CXXC zinc finger domain. BAH: Bromo-Adjacent Homology domain (BAH1/2). PBHD: Poly Bromo Homology Domain. PWWP: Pro-Trp-Pro-Trp domain. ADD: atrx-DNMT3-DNMT3L domain. KG repeats: Lysine–Glycine dipeptide repeats. TRD: Target Recognition Domain. CFT: CFTXXYXXY motif.

**Figure 4 ijms-26-10496-f004:**
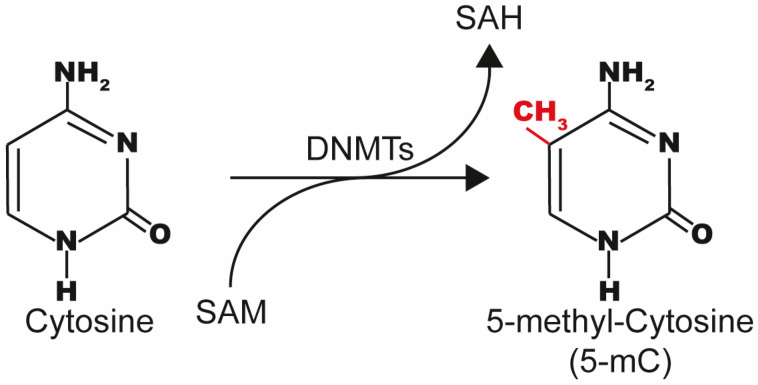
The mechanism of DNA methylation via DNMT enzymes. DNMT enzymes add a methyl group from SAM to the carbon in the 5-position in cytosine in the CpG context on DNA. Methylated cytosines are shown in red.

**Table 2 ijms-26-10496-t002:** The diagnostic value of DNA methylation in CC.

Gene or DNA Region	Assay	Results	Study Population	Cite
*ZNF671*	GynTect^®^	Methylation in its promoter exhibited the highest AUC of 0.811 (95% C.I.: 0.734–0.888) for identifying CIN3+ cases with positive HR-HPV DNA	38 and 61 normal and CC cytology samples from Chinese women	[182]
*ASTN1* *DLX1* *ITGA4* *RXFP3* *SOX17* *ZNF671*	GynTect^®^	The methylation of these 6 genes showed a sensitivity of 88.5% and specificity of 87.2% in the triage of HPV-positive women (CIN3+)	51 CC LBC and 45 CIN3 LBC samples of women from different regions of China	[183]
*CADM1* *MAL*	ddPCR	Methylation in their promoters presented an AUC = 0.912, 100% specificity, and 70% sensitivity	12 CC biopsies samples of HPV-positive Russian women	[187]
*miR-124-2* *MAL*	ddMSP	Identification of methylation in their promoters: positivity in 73.17% samples, with 73.71 (95% C.I.: 59.61–86.73) sensitivity and 77.23% (95% C.I.: 74.11–80.35) specificity	41 CIN3+ scrape (HPV16/33 positives) samples from Northern Portuguese woman	[188]
*SOX1*	qMSP/BSP/Pyrosequencing	Diagnostic value of hypermethylation in its promoter was 68.95 (95% C.I.: 27.63–172.07), with an AUC of 0.92	1024 CC tissue and 2189 exfoliated cells from different populations worldwide	[189]
*ITGA4*	HT sequencing	ITGA4 methylation showed an AUC of 0.866 (95% C.I.: 0.806–0.925), specificity of 96.45% (95% C.I.: 0.936–0.981), and sensitivity of 75.32% (95% C.I.: 0.647–0.836) for detecting CIN2+ patients with HR-HPV infection	57 CIN2, 7 CIN3, and 14 CC smear samples from Chinese women	[190]
*ZNF671*	qMSP*	Methylation in this gene promoter presented an AUC of 0.945, sensitivity of 91.5%, and specificity of 92.4%	21 HSIL and 47 CC smear samples from China	[184]
*PAX1* *NREP-AS1*	850k EPIC methylation array	PAX1 and NREP-AS1 methylation showed an AUC of 0.77, with a sensitivity of 83% and a specificity of 87% for CIN3+ patients	73 CIN3 and 54 CC LBC samples from UK and Greek women	[191]
*PAX1* *JAM3*	CISCER^®^	Methylation in these gene promoters revealed a sensitivity of 74.9% (95% CI, 68.3–81.4%) and a specificity of 89.1% (95% CI 87.6–90.6%) for detecting CIN3+	149 CIN3 and 18 CC tissue samples from China	[192]
*PAX1* *JAM3*	PAX1 and JAM3 gene methylation detection kit	Methylation in *PAX1* and *JAM3* promoters was detected in all CC patients and in 89.6% of CIN3+ cases, with an AUC of 0.790 (95% CI, 0.747–0.832)	95 CIN3 and 23 CC LBC samples from China	[193]
*ZSCAN18*	Pyrosequencing/qMSP/MSP	*ZSCAN18* promoter methylation presented an AUC of 0.9421	167 normal and 596 CC tissue samples from TCGA, GEO databases, and China	[185]
*METloc001* *METloc002*	qMSP	Identification of methylation: positivity in 37.1% samples, with 93.8 (95% C.I.: 69.8–99.8) sensitivity and 75.3% (95% C.I.: 63.9–84.7) specificity	89 CIN3+ LBC samples from Danish women aged 45+ with a TZ3	[194]
*FAM19A4* *miR-124-2*	QIAsure^®^	Methylation observed in 100% of CC cases	6 CC LBC samples from Greek women	[186]

CIN3+: Cervical Intraepithelial Neoplasia Grade 3 lesions or worse; CIN2+: Cervical Intraepithelial Neoplasia Grade 3 lesions or worse; ddPCR: droplet digital PCR; AUC: Area Under the Curve; ddMSP: droplet digital methylation-specific PCR; LBC: liquid-based cytology; TZ3: type 3 transformation zone; qMSP*: quantitative multiplex methylation-specific PCR; qMSP: multiplex quantitative methylation-specific PCR; MSP: methylation-specific PCR; BSP: Bisulfite Sequencing PCR; TCGA: The Cancer Genome Atlas; HT sequencing: high-throughput sequencing.

**Table 3 ijms-26-10496-t003:** The prognostic value of DNA methylation in CC.

Gene/DNA Region	Assay	Results	Study Population	Cite
*ASTN1* *DLX1* *ITGA4* *RXFP3* *SOX17* *ZNF671*	GynTect^®^	GynTect^®^ showed an AUC of 0.715 (95% C.I.: 0.592–0.837), with a specificity of 70.4% and sensitivity of 72.5% in predicting postoperative specimen histology	28 CIN2 and 50 CIN3 tissue samples from Chinese women	[195]
*PRMD8* *MIR520H*	Infinium MethylationEPIC BeadChip array	A 24 CpGs signature with aberrant methylation was identified, including hypermethylation in tumor-suppressor genes as *PRMD8* and hypomethylation in oncogenes such as *MIR520H*. A total of 92.9% sensitivity and 88.6% specificity was observed in predicting CC risk	A total of 538 tissue samples were analyzed, including CC and HIV-positive Nigerian women	[196]
*ASCL1* *LHX8* *GHSR* *SST* *ZIC1*	qMSP and precursor-M Gold methylation assay	DNA methylation positivity in 77.8% of patients with recurrence	Cervicovaginal samples of 47 patients without recurrence and 20 patients with recurrence from the Netherlands	[197]

qMSP: multiplex quantitative methylation-specific PCR.

## Data Availability

No new data were created or analyzed in this study. Data sharing is not applicable to this article.

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
