# Peer review of "DNMT Enzymes and Their Impact on Cervical Cancer: A State-of-the-Art Review"

_ijms, 2025, doi:10.3390/ijms262110496_

Round 1

Reviewer 1 Report

Comments and Suggestions for Authors

The review is thoroughly written and well-structured, describing DNMT enzymes, which play a crucial role in epigenetic modification. It provides an overview of their discovery, structure, function, and association with cervical cancer, as well as a summary of medications that target DNMT enzymes.

However i did find some minor errors.

Comments on the Quality of English Language

  • Line 58, 60: Ensure 5mC is written consistently as (5-mC) throughout the text.

  • Line 62: Cytosine should not be capitalized.

  • Line 63: Methyltransferases should not be capitalized.

  • Line 67: Replace is high with is increased for improved scientific precision.

  • Lines 69–74: Please add an appropriate reference to support this statement.

  • Line 83: Clarify that incidence and mortality values refer to rates per 100,000 women per year.

  • Line 106: Adjust word order to: the most studied epigenetic mechanism in cervical cancer (CC) is…

  • Line 116: Cytosines should not be capitalized.

  • Line 123: Replace after with afterward for correct usage.

  • Line 148: Bladder, kidney, colon, and pancreas should not be capitalized.

  • Line 153: Replace primary acute cell with primary acute leukemia cell.

  • Line 195: Havana should appear in all capitals (HAVANA).

  • Line 196: Remove the unnecessary word for.

  • Line 210: Revise to: an mRNA of 4,336 bp from the forward strand (Figure 2D, lower) that encodes a protein of 853 amino acids (95.751 kDa).

  • Line 217: Remove the unnecessary for.

  • Line 632: Replace was the more assay used with was the most frequently used assay.

  • Line 632: Replace for diagnostic of with for the diagnosis of.

  • Line 638: Replace to with for.

  • Line 645: Replace prognostic with prognosis.

  • Line 647: Delete using a larger sample size and change it with  larger cohorts.

  • Line 694: Replace his with This.

  • Final note: Paragraph numbering from section 13. Future Directions onward is incorrect and should be revised.

Author Response

Author response to reviewers’ comments on: “DNMT Enzymes and Their Impact on Cervical Cancer: A State-of-the-Art Review” by Eric Genaro Salmerón-Bárcenas…….et al.,

We thank the reviewer 1 for their fair and critical assessment of our manuscript.

The edition of the new version of the manuscript entitled has been done. The changes made in the manuscript are indicated in red letters. Please find our point-by-point responses below,

Q1. Line 58, 60: Ensure 5mC is written consistently as (5-mC) throughout the text.

Answer 1. The correction has been made, and 5mC has been consistently replaced with 5-mC throughout the text.

Q2. Line 62: Cytosine should not be capitalized.

Answer 2. Thank you so much for your comments. Currently, “cytosine” is written in lowercase. Please refer to line 62

Q3 Line 63: Methyltransferases should not be capitalized.

Currently, methyltransferases are written in lowercase. Please check this line 63.

Q4 Line 67: Replace is high with is increased for improved scientific precision.

Answer 4. Thank you for your help. The revised sentence is:

“Expression of DNMT enzymes is increased in several cellular processes, such as DNA replication, as well as in several human diseases, including cancer [19, 20]”. Lines 67-68.

Q5 69–74: Please add an appropriate reference to support this statement.

The following reference has been included:

Bray, F.; Laversanne, M.; Sung, H.; Ferlay, J.; Siegel, R.L.; Soerjomataram, I.; Jemal, A. Global cancer statistics 2022: GLOBOCAN estimates of incidence and mortality worldwide for 36 cancers in 185 countries. CA: A Cancer Journal for Clinicians 2024, 74, 229–263.

This reference supports the statement and corresponds to reference 21, lines 784-786.

Q6 83: Clarify that incidence and mortality values refer to rates per 100,000 women per year.

Answer 6. The incidence and mortality values have been clarified as follows:

“In addition, the regions of each continent with the highest incidence and mortality rates of CC are Eastern Africa (incidence: 40.4 and mortality: 28.9 per 100,000 women per year), South-Eastern Asia (incidence: 17.4 and mortality: 9.5 per 100,000 women per year), Eastern Europe (incidence: 15.7 and mortality: 6.3 per 100,000 women per year), and South America (incidence: 15.6 and mortality: 7.8 per 100,000 women per year) [21].” Lines 81-86.

Q7 106: Adjust word order to: the most studied epigenetic mechanism in cervical cancer (CC) is…

Answer 7. The word order has been adjusted as follows: “The most epigenetic mechanisms studied in CC is the DNA methylation mediated by DNMT enzymes [35, 36]”. Lines 109-110.

Q8 116: Cytosines should not be capitalized.

Answer 8. Thank you so much for your comments. Currently, “cytosines” is written in lowercase. Please refer to line 119.

Q9. Line 123: Replace after with afterward for correct usage.

Answer Q9. The Replace has been made and the new sentences is:

“Afterward, in 1988, it was cloned, purified, and characterized the first mammalian DNMT enzyme, called DNMT1, from Murine erythroleukemia  cells [43]”. Lines 126-127.

Q10. Line 148: Bladder, kidney, colon, and pancreas should not be capitalized.

Answer 10. Bladder, kidney, colon, and pancreas  have been written correctly. The revised sentence is:

Moreover, overexpression of DNMT1, DNMT3A, and DNMT3B at the mRNA level was observed in bladder, kidney, colon, and pancreas human cancers [51].  Lines 149-150.

Q11. Line 153: Replace primary acute cell with primary acute leukemia cell.

Answer 11.The correction has been made, and primary acute cell has been consistently replaced with” primary acute leukemia cell” throughout the text. Please see the corresponding lines 154- 156.

Q12. Line 195: Havana should appear in all capitals (HAVANA).

Answer 12. Havana is now written in capital letters. Lines 187, 198, 218, 228

Q13 Line 196: Remove the unnecessary word for.

Answer 13. The world for has been removed as you can see in the follows sentence:
“A total of 11 new transcripts were annotated in Ensembl and HAVANA databases, including four transcripts with potential to code for proteins, three transcripts with nonsense-mediated decay, and four transcripts whose CDS are not yet defined [56].”(Lines 197- 200).

Q14. Line 210: Revise to: an mRNA of 4,336 bp from the forward strand (Figure 2D, lower) that encodes a protein of 853 amino acids (95.751 kDa).

Answer 14. The sentence was revised, and the new sentence is:
“The DNMT3B gene transcribes a 4,336 bp mRNA from the forward strand (Figure 2D, lower), which encodes a protein of 853 amino acids (95.751 kDa).” (ENST00000328111.6/hDNMT3B-202)”
lines 213-214

Q15. 217: Remove the unnecessary for.

Answer 15. The unnecessary world for has been removed as you can see in the follows sentence:

Specifically, seven transcripts were annotated with the potential to code proteins, five transcripts with nonsense-mediated decay, two transcripts classified as protein-coding CDS not defined, and six transcripts considered as retained introns [56]. Lines 219-222.

Q16. 632: Replace was the more assay used with was the most frequently used assay.

Answer 16. The replacement has been made. The revised sentence is:
“Interestingly, the GynTect® test was the most frequently used assay this year to diagnose women with CC.” Lines 645-646

Q17. 632: Replace for diagnostic of with for the diagnosis of.

Answer 17. The replacement has been made. The new sentences is:

Interestingly, the GynTect® test was the most frequently used assay this year to diagnose women with CC.”. Lines 645-646

Q18. 638: Replace to with for.

Answer 18. The replacement has been made as follows:

“Surprisingly, a study reported an AUC of 0.9421 for methylation in the ZSCAN18 promoter using three different assays and a total of 763 samples.” (Lines 652-654)

Q19.  645: Replace prognostic with prognosis.

Answer 19. The replacement has been made. Please see line […] with the current sentence: “In addition, in 2025, some studies demonstrated the potential use of methylation biomarkers for the prognosis of CC patients, employing techniques such as GynTect® Arrays and qMSP (Table 3)” Lines 658-660.

Q20. 647: Delete using a larger sample size and change it with  larger cohorts.

Answer 20. The new sentence is as follows:
“However, all these studies need to be further validated in independent cohorts with other techniques and larger cohorts.” Lines 660-661

Q21.  694: Replace his with This.

Answer 21. Sorry for this unintentional mistake. The correction has been made as follows:
This opens the door to exploring whether DNMT enzymes—or their catalytic domains—could be used as targeted molecular tools to repress.” (Lines  707-708)

Q22. Final note: Paragraph numbering from section 13. Future Directions onward is incorrect and should be revised.

Answer 22 Answer 22: After careful review, we did not identify any issues with the paragraph numbering in this section.  

Reviewer 2 Report

Comments and Suggestions for Authors

This manuscript by Salmerón-Bárcenas et al. describes the critical roles of DNA methyltransferases (DNMTs) in cervical cancer, emphasizing their involvement in aberrant DNA methylation and gene silencing. The authors summarize the role of DNMT expression and regulation in cervical carcinogenesis, and discuss their potential as diagnostic, prognostic, and therapeutic biomarkers. The topic is of interest to readers. I recommend it for publication after addressing several minor points.

  1. Some relevant studies regarding the role of DNMTs in cervical cancer should be cited. For example, a recent study demonstrated that the DNA mismatch repair system regulates PD-L1 expression through DNMTs in cervical cancer. Please include such related references.
  2. In Figure 1, the chemical structure of 5-mC should be corrected.
  3. Similarly, please revise the chemical structures of cytosine and 5-mC (NH₂ and CH₃ groups). In addition, the atom numbering should be shown as subscripts.
  4. It is recommended to include additional DNMT inhibitors reported to target cervical cancer beyond those listed in Table 1.
  5. Please add an “Abbreviations” section at the end of the manuscript.
  6. For the reference section, please unify the journal name formatting (e.g., abbreviations, as in reference 33).

Author Response

Reviewer 2

Author response to reviewers’ comments on: “DNMT Enzymes and Their Impact on Cervical Cancer: A State-of-the-Art Review” by Eric Genaro Salmerón-Bárcenas…….et al.,

We thank the reviewer 2 for their fair and critical assessment of our manuscript.

The edition of the new version of the manuscript entitled has been done. The changes made in the manuscript are indicated in red letters. Please find our point-by-point responses below,

Q1 Some relevant studies regarding the role of DNMTs in cervical cancer should be cited. For example, a recent study demonstrated that the DNA mismatch repair system regulates PD-L1 expression through DNMTs in cervical cancer. Please include such related references.

Answer 1. Two additional articles about relevant studies regarding the role of DNMTs in cervical cancer were added:

  1. Kim, M. J.; Lee, H. J.; Choi, M. Y.; Kang, S. S.; Kim, Y. S.; Shin, J. K.; Choi, W. S. UHRF1 Induces Methylation of the TXNIP Promoter and Down-Regulates Gene Expression in Cervical Cancer. Mol Cells 2021, 44, 146–159. DOI: 10.14348/molcells.2021.0001. (Please see lines 337–338)
  2. Li, L.; Xu, C.; Long, J.; Shen, D.; Zhou, W.; Zhou, Q.; Yang, J.; Jiang, M. E6 and E7 Gene Silencing Results in Decreased Methylation of Tumor Suppressor Genes and Induces Phenotype Transformation of Human Cervical Carcinoma Cell Lines. Oncotarget 2015, 6, 23930–23943. DOI: 10.18632/oncotarget.4525. (Lines 358–360)

However, these articles were not included in this section but rather in the corresponding section for each type of DNMT.

Q2. In Figure 1, the chemical structure of 5-mC should be corrected.

Answer 2. Thank you very much for your valuable comments. The chemical structure has been corrected accordingly. The correct structure is now shown below and can be found in the lines 156-157.

Q3. Similarly, please revise the chemical structures of cytosine and 5-mC (NH₂ and CH₃ groups). In addition, the atom numbering should be shown as subscripts.

Thank you very much for your valuable comment. The chemical structure of cytosine has been revised to accurately include the NH₂ and CH₃ groups with their respective subscripts. These corrections ensure the proper representation of the molecule. The updated structure can be seen in the lines 156-157 (figure 1) and lines 289-290 (Figure 4).

Q4. It is recommended to include additional DNMT inhibitors reported to target cervical cancer beyond those listed in Table 1.

Eight additional DNMT inhibitors were included in Table 1, as indicated in red letter (References 161-168)

Q5. Please add an “Abbreviations” section at the end of the manuscript.

Answer 5. In response to the reviewer’s comment, the abbreviation section has been added to the manuscript and can be found in lines 734-736

Q6. For the reference section, please unify the journal name formatting (e.g., abbreviations, as in reference 33).

Answer 6. The references have been updated to include the abbreviated names of the journals in the References section, as indicated in the lines 736-1256.